# STABILIZING THE KUMARASWAMY DISTRIBUTION

## ABSTRACT

Large-scale latent variable models require expressive continuous distributions that support efficient sampling and low-variance differentiation, achievable through the reparameterization trick. The Kumaraswamy (KS) distribution is both expressive and supports the reparameterization trick with a simple closed-form inverse CDF. Yet, its adoption remains limited. We identify and resolve numerical instabilities in the inverse CDF and log-pdf, exposing issues in libraries like PyTorch and TensorFlow. We then introduce simple and scalable latent variable models to improve exploration-exploitation trade-offs in contextual multi-armed bandits and enhance uncertainty quantification for link prediction with graph neural networks. We find these models to be most performant when paired with the stable KS. Our results support the stabilized KS distribution as a core component in scalable variational models for bounded latent variables.

## 1 INTRODUCTION

Probabilistic models use probability distributions as building blocks to model complex joint distributions between random variables. Such distributions can model unobserved 'latent' variables $z$, or observed 'data' variables $x$. Bounded interval-supported latent variables are central to many key applications, such as unobserved probabilities (e.g., user clicks in recommendation systems or links between network nodes), missing measurements in control systems (e.g., joint angles in $[0, 2\pi]$), and stochastic policies over bounded actions in reinforcement learning (e.g., motor torque in $[-10, 10]$).

To meet the demands of large-scale latent variable models, bounded interval-supported distributions must satisfy the following criteria: (i) support the reparameterization trick through an explicit reparameterization function, such as a closed-form inverse CDF, enabling efficient sampling and low-variance gradient estimates; (ii) provide sufficient expressiveness to capture complex latent spaces; and (iii) offer simple distribution-related functions (log-pdf, explicit reparameterization function, and gradients) that allow fast and accurate evaluation. In Section 2, we argue that the Kumaraswamy (KS) distribution uniquely meets these criteria, yet remains surprisingly underused.

In this paper, we make the following technical contributions:

- We identify and resolve numerical instabilities in the KS's log-pdf and inverse CDF, impacting core auto-differentiation libraries. To this end, we introduce an unconstrained logarithmic parameterization, enhancing its compatibility with neural network (NN) settings (Section 3).

- We propose the Variational Bandit Encoder (VBE), addressing exploration-exploitation trade-offs in contextual Bernoulli multi-armed bandits (Section 4.2).

- We propose the Variational Edge Encoder (VEE) for improved uncertainty quantification in link prediction with graph neural networks (Section 4.3).

With the stabilized KS distribution at their core, these simple and scalable variational models open new avenues for addressing pressing challenges in large-scale latent variable models, including those in recommendation systems, reinforcement learning, and network analysis.

## 2 BACKGROUND

The KS distribution (Kumaraswamy, 1980; Jones, 2009) has pdf $f(x) = abx^{a-1}(1-x^a)^{b-1}$ and inverse CDF $F^{-1}(u) = (1-u^{b^{-1}})^{a^{-1}}$, both defined for $x, u \in (0, 1)$ and parameterized by $a, b > 0$.

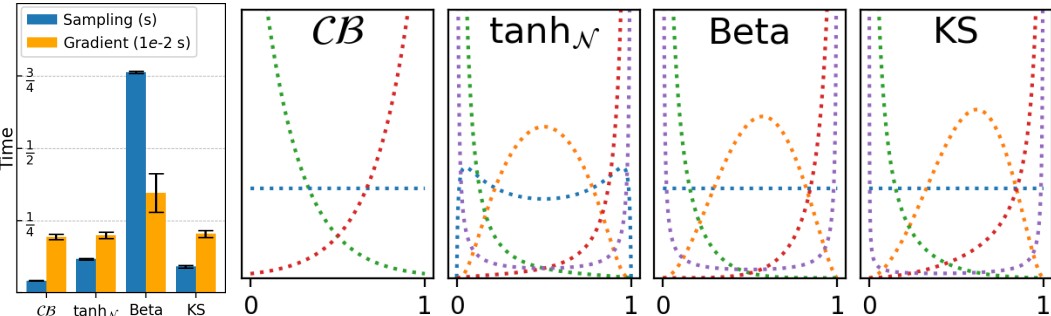

Figure 1: Comparison of relevant bounded interval-supported distributions. Left: Time for sampling and differentiating through samples. The Beta lacks explicit reparameterization, and has slower sampling and gradients. Right: Expressiveness in terms of attainable prototypical shapes.

**Continuous distributions with bounded interval support.** Among bounded interval-supported distributions, the KS uniquely satisfies criteria (i)–(iii) in Section 1. It supports the reparameterization trick through its closed-form, differentiable inverse CDF, providing efficient sampling and low-variance gradients. The KS supports four distinct prototypical shapes — bell, U, increasing, and decreasing (Figure 1, right) — providing expressivity for diverse modeling tasks. Its log-pdf and inverse CDF, along with their gradients, are composed only of affine transformations, exponentials, and logarithms, and can be parameterized directly in terms of unconstrained logarithmic values. This enables straightforward implementation with minimal dependencies and keeps most computation in the more stable and accurate log-space. The unconstrained logarithmic parameterization makes it well-suited for NNs, eliminating the need for positivity-enforcing link functions. Additionally, the KS has differentiable, closed-form expressions for moments, entropy $\mathcal{H}$, and the Kullback-Leibler (KL) divergence to the Beta distribution, facilitating efficient incorporation of prior information.

Other common interval-supported distributions face limitations. The Continuous Bernoulli ($\mathcal{CB}$) distribution is less expressive with only a single parameter. Truncated distributions lack the reparameterization trick and often require slower, rejection-based sampling methods (Figurnov et al., 2018). Squashed Gaussian distributions, like the tanh-normal ($\tanh_{\mathcal{N}}$), support the reparameterization trick but cannot represent the uniform distribution — limiting it's ability to accurately capture uncertainty — and suffer from numerical instabilities in the log-pdf and require sample approximations for moments, entropy, and KL divergences to distributions outside their family. Mitigating log-pdf instability typically requires careful control of the underlying Gaussian parameters and clipping of log-pdf values (Haarnoja et al., 2018). The two-parameter Beta distribution shares the same four fundamental shapes as the KS and benefits from being in the exponential family, which provides a rich set of KL divergences. However, it lacks the reparameterization trick, relying on rejection sampling for generation and implicit reparameterization (reviewed later in this section) for gradient computation. Figure 1 (left) shows Beta sampling is nearly an order of magnitude slower than KS sampling on an Apple M2 CPU. See Appendix A.1 for more distributional comparisons.

**Latent variable modeling with stochastic variational inference (SVI).** The primary method for fitting large-scale latent variable models is SVI (Hoffman et al., 2013). Consider a model $p_{\boldsymbol{\theta}}(\boldsymbol{x}) = \int p_{\boldsymbol{\theta}}(\boldsymbol{x}|\boldsymbol{z})p(\boldsymbol{z})d\boldsymbol{z}$, where $\boldsymbol{x} \in \mathbb{R}^M$ is the observation, $\boldsymbol{z} \in \mathbb{R}^D$ is a vector-valued latent variable, $p_{\boldsymbol{\theta}}(\boldsymbol{x}|\boldsymbol{z})$ is the likelihood function with parameters $\boldsymbol{\theta}$, and $p(\boldsymbol{z})$ is the prior distribution. Except for a few special cases, maximum likelihood learning in such models is intractable because of the difficulty of the integrals involved. Variational inference (Jaakkola & Jordan, 2000) provides a tractable alternative by introducing a variational posterior distribution $q_{\boldsymbol{\phi}}(\boldsymbol{z}|\boldsymbol{x})$ and maximizing a lower bound on the marginal log-likelihood called the ELBO:

$$\mathcal{L}(\boldsymbol{x}, \boldsymbol{\theta}, \boldsymbol{\phi}) = \mathbb{E}_{q_{\boldsymbol{\phi}}(\boldsymbol{z}|\boldsymbol{x})}\left[\log p_{\boldsymbol{\theta}}(\boldsymbol{x}|\boldsymbol{z})\right] - D_{\mathrm{KL}}\left(q_{\boldsymbol{\phi}}(\boldsymbol{z}|\boldsymbol{x}) \,\|\, p(\boldsymbol{z})\right) \leq \log p_{\boldsymbol{\theta}}(\boldsymbol{x}). \quad (1)$$

Training models with modern SVI (Kingma & Welling, 2014; Rezende et al., 2014) involves gradient-based optimization of this bound w.r.t. both the model parameters $\boldsymbol{\theta}$ and the variational parameters $\boldsymbol{\phi}$. The first term in (1) encourages the model to assign high likelihood to the data, but its exact evaluation and gradients are typically intractable and so the expectation is often approximated with samples from $q_{\boldsymbol{\phi}}(\boldsymbol{z}|\boldsymbol{x})$. The KL divergence term incorporates prior information by

penalizing deviations of the variational posterior from the prior $p(\boldsymbol{z})$. Closed-form expressions of $D_{\mathrm{KL}}(q_{\boldsymbol{\phi}}(\boldsymbol{z}|\boldsymbol{x}) \| p(\boldsymbol{z}))$ allow efficient encoding of prior information; otherwise, sample-based approximations are required. Modifying the ELBO by scaling the KL term with a parameter $\beta_{\mathrm{KL}} > 0$ is often necessary to balance the trade-off between data likelihood and prior regularization (Alemi et al., 2018). We denote the sample-based approximation of this modified ELBO as $\hat{\mathcal{L}}_{\beta_{\mathrm{KL}}}$.

**Gradient reparameterization: explicit and implicit.** A distribution $q_{\boldsymbol{\phi}}(\boldsymbol{z})$ is said to be *explicitly* reparameterizable, or amenable to the 'reparameterization trick', if it can be expressed as a deterministic, differentiable transformation $\boldsymbol{z} = g(\boldsymbol{\epsilon}, \boldsymbol{\phi})$ of a base distribution $\boldsymbol{\epsilon} \sim p(\boldsymbol{\epsilon})$. This base distribution is typically simple, such as Uniform or standard Normal, enabling fast sample generation by first sampling from the base and then applying $g$. This enables the use of backpropagation to compute gradients of the form [cf. (1)]

$$\nabla_{\boldsymbol{\phi}}\mathbb{E}_{q_{\boldsymbol{\phi}}(\boldsymbol{z})}[f(\boldsymbol{z})] = \mathbb{E}_{p(\boldsymbol{\epsilon})}[\nabla_{\boldsymbol{\phi}}f(g(\boldsymbol{\epsilon}, \boldsymbol{\phi}))] = \mathbb{E}_{p(\boldsymbol{\epsilon})}[\nabla_{\boldsymbol{z}}f(\boldsymbol{z})|_{\boldsymbol{z}=g(\boldsymbol{\epsilon}, \boldsymbol{\phi})}\nabla_{\boldsymbol{\phi}}g(\boldsymbol{\epsilon}, \boldsymbol{\phi})], \qquad (2)$$

an expectation with form encompassing the ELBO. Explicit reparameterization is compatible with distributions in the location-scale family (e.g., Gaussian, Laplace, Cauchy), distributions with tractable inverse CDFs (e.g., exponential, KS, $\mathcal{CB}$), or those expressible as deterministic transformations of such distributions (e.g., $\tanh_{\mathcal{N}}$). When explicit reparameterization is not available, implicit reparameterization (Figurnov et al., 2018) is commonly used for distributions with numerically tractable CDFs, such as truncated, mixture, Gamma, Beta, Dirichlet, or von Mises distributions. This method expresses the parameter gradient through the sample $\nabla_{\boldsymbol{\phi}}\boldsymbol{z}$ as a function only of the CDF gradients, not its inverse. Such CDF gradients are either found analytically (if feasible) or more commonly using numerical methods, e.g., forward mode auto-differentiation on CDF estimates, as in the Gamma and Beta distributions. Without explicit reparameterization, sampling and gradient computations tend to be slower and more complex, and produce higher-variance estimates of (2), reducing learning efficiency and stability (Kingma & Welling, 2014; Jang et al., 2017).

## 3 STABILIZING THE KUMARASWAMY

**Identifying the instability**: $\log(1 - \exp(x))$. Naive computation of $\log(1 - \exp(x))$ for $x < 0$ leads to significant numerical errors as $x$ approaches 0 (Figure 2, red). These errors grow so large that they can cause *numerical instability*, i.e., an irrecoverable error such as $-\texttt{inf}$. These errors result from *catastrophic cancellation*, which occurs when subtracting nearly equal numbers — here, $1 - \exp(x)$. As $x \to 0$, $\exp(x) \approx 1$, so $\texttt{1 - exp(x)}$ results in the cancellation of leading significant bits, leaving only a few less significant, less accurate bits to represent the result. This causes large relative errors in $\texttt{1 - exp(x)}$, which are amplified when input to the logarithm as its magnitude grows sharply near zero. If the cancellation is complete, $\texttt{1 - exp(x)}$ underflows to 0 and the logarithm returns $-\texttt{inf}$, as seen in Figure 2 (red) when $\log_2 |x| < -24$.

When $x \approx 0$, $\log(1 + x)$ and $\exp(x) - 1$ can be accurately computed using Taylor series expansions, implemented as $\texttt{log1p}$ and $\texttt{expm1}$, respectively (see Appendix A.2). These functions form the basis for two common methods to compute $\log(1 - \exp(x))$: $\texttt{log(-expm1(x))}$ and $\texttt{log1p(-exp(x))}$. (Mächler, 2012) showed neither method provides sufficient accuracy across the domain. However, each approach is accurate in complementary regions, leading to

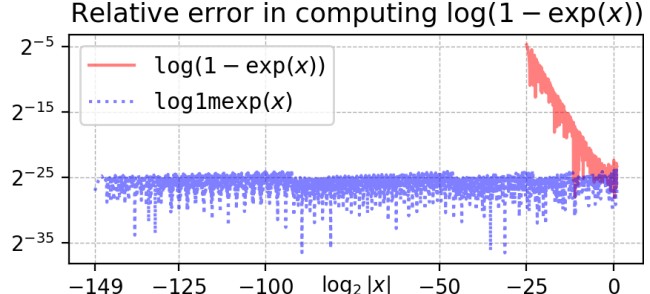

Figure 2: Naive computation of $\log(1 - \exp(x))$ (red) becomes unstable as $x \to 0$ due to catastrophic cancellation, while $\texttt{log1mexp(x)}$ (blue) ensures accurate computation.

$$\texttt{log1mexp}(x) := \begin{cases} \texttt{log(-expm1(x))} & -\log 2 \leq x < 0 \\ \texttt{log1p(-exp(x))} & x < -\log 2, \end{cases} \qquad (3)$$

which computes $\log(1 - \exp(x))$ accurately throughout single precision, shown in Figure 2 (blue).

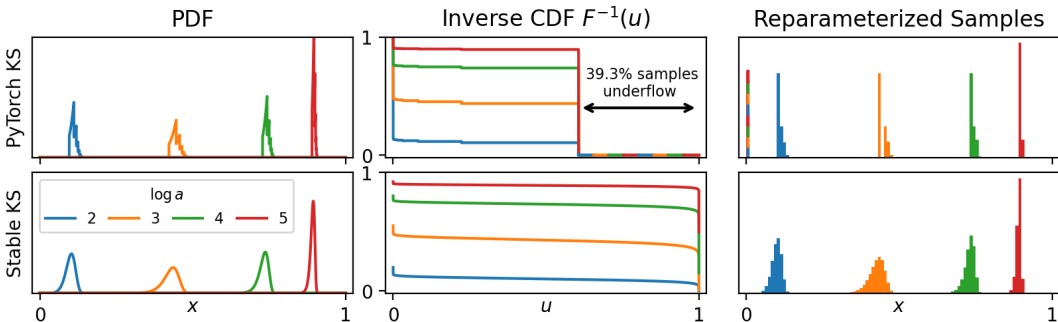

Figure 3: Stabilizing $\log(1 - \exp(x))$ terms eliminates numerical instabilities in the KS log-pdf and inverse CDF. We compare the unstable PyTorch KS implementation (top row) and our stable KS (bottom row) for realistic KS distributions ($\log_2 b = 24$, varying $a$). Catastrophic cancellation in the $\log(1 - \exp(x))$ terms in the PyTorch KS causes jagged curves and inverse CDF underflow beyond $u \approx 1 - 39.3$, resulting in a point mass of $\approx 39.3$ at $x = 0$ in the sampling distribution. Our stable KS removes the instability by using `log1mexp`.

**A Stable Kumaraswamy.** The direct implementation of the KS's log-pdf and inverse CDF — as found in all core auto-differentiation libraries — produces numerical instabilities. Here, we introduce a novel parameterization in terms of unconstrained logarithmic parameter values, which isolates and makes explicit the unstable terms

$$w_{b^{-1}}(u) = \log(1 - u^{b^{-1}}) = \log(1 - \exp(b^{-1}\log u))$$
$$w_a(x) = \log(1 - x^a) \quad = \log(1 - \exp(a\log x)),$$

eliminates the need for positivity-enforcing link functions, and whose expressions involve only affine, exponential, and logarithmic transformations. This allows the log-pdf, inverse CDF, and their gradients to be expressed as:

$$\log f(x) = \log a + \log b + (a - 1)\log x + (b - 1)w_a(x) \tag{4}$$
$$\nabla_{\log x}\log f(x) = (a - 1) - (b - 1)\cdot\exp(a\log x - w_a(x) + \log a) \tag{5}$$
$$\nabla_{\log a}\log f(x) = 1 + a\log x\cdot\{1 - (b - 1)\cdot\exp(a\log x - w_a(x))\} \tag{6}$$
$$\nabla_{\log b}\log f(x) = 1 + b\cdot w_a(x) \tag{7}$$

$$F^{-1}(u) = (1 - u^{b^{-1}})^{a^{-1}} = \exp(a^{-1}w_{b^{-1}}(u)) \tag{8}$$
$$\nabla_{\log a}F^{-1}(u) = \exp(-\log a + a^{-1}w_{b^{-1}}(u))\cdot(-w_{b^{-1}}(u)) \tag{9}$$
$$\nabla_{\log b}F^{-1}(u) = \exp(-\log a - \log b + b^{-1}\log u + (a^{-1} - 1)w_{b^{-1}}(u))\cdot\log u. \tag{10}$$

This parameterization's algebraic form allows direct replacement of the dominant unstable terms, substituting $w_{b^{-1}}(u)$ with `log1mexp`$(b^{-1}\log u)$ and $w_a(x)$ with `log1mexp`$(a\log x)$. Access to $\log a$ and $\log b$ avoids errors from unnecessary transitions in-and-out of log-space. We also avoid the error prone expressions produced in backpropogation's direct application of the chain rule, e.g., $\frac{1}{a}\cdot\exp\left(\frac{1}{a}\log\left(1\text{-}\exp\left(\frac{1}{b}\log u\right)\right)\right)\cdot\text{-}\left(1\text{-}\exp\left(\frac{1}{b}\log u\right)\right)^{-1}\cdot\exp\left(\frac{1}{b}\log u\right)\cdot\log u\cdot\frac{-1}{b^2}\cdot b$ and (10) are equivalent expressions for $\nabla_{\log b}F^{-1}$, but their computed values can differ greatly for extreme parameter values. Desirable KS distributions can obtain such problematic extreme parameter values, e.g., the KS distributions in Figure 3 have $b \approx 10^6$. See Appendix A.3 for further discussion on how instability in the unmodified KS can worsen with increasing evidence.

Figure 3 compares the PDF, inverse CDF, and histograms of reparameterized samples for KS distributions which are typical to real-world modeling scenarios. The PyTorch implementation (top row) shows jaggedness in both the PDF and inverse CDF, caused by catastrophic cancellation in the unstable terms $w_a(x)$ and $w_{b^{-1}}(u)$. Additionally, the PyTorch inverse CDF underflows beyond $u \approx 1 - 39.3$: here, $w_{b^{-1}}(u) = -\infty$, and $F^{-1}(u) = \exp(a^{-1}\cdot-\infty) = 0$. This underflow results in a point mass at $x = 0$ (a point outside of the KS support) with probability $\approx 39.3$ in each of the reparameterized sampling distributions, and produces infinite gradients via $\nabla_{\log a}F^{-1} = \infty$ [cf. (9)]. This infinite gradient triggers a cascade: infinite parameter values after the optimizer step and `NaN` activations in the next forward pass, which is what users ultimately observe when training fails.

Table 1: VAE on MNIST and CIFAR-10.

| Prior | $q_{\phi}(\boldsymbol{z}|\boldsymbol{x})$ | $p_{\theta}(\boldsymbol{x}|\boldsymbol{z})$ | MNIST | | CIFAR-10 | |
|---|---|---|---|---|---|---|
| | | | ELBO | $\mathcal{K}(\phi)$ | ELBO | $\mathcal{K}(\phi)$ |
| $\mathcal{N}_{(0,1)}$ | $\mathcal{N}$ | $\mathcal{CB}$ | **1825** | 97.3 | 1167 | 37.9 |
| $U_{(0,1)}$ | KS | $\mathcal{CB}$ | 1818 | 97.4 | **1172** | **41.5** |
| $U_{(0,1)}$ | Beta | $\mathcal{CB}$ | 1821 | **97.5** | 1167 | 40.3 |
| $\mathcal{N}_{(0,1)}$ | $\mathcal{N}$ | Beta | 4073 | **92.1** | **3566** | 48.5 |
| $U_{(0,1)}$ | KS | Beta | 4061 | 91.3 | 3483 | **50.1** |
| $U_{(0,1)}$ | Beta | Beta | **4082** | 90.1 | N/A | N/A |
| $\mathcal{N}_{(0,1)}$ | $\mathcal{N}$ | KS | 3328 | 96.4 | 1720 | 47.1 |
| $U_{(0,1)}$ | KS | KS | **3355** | 96.8 | **1738** | 48.8 |
| $U_{(0,1)}$ | Beta | KS | 3348 | **97.1** | N/A | N/A |

Table 2: MNIST test digit VAE reconstructions.

## 4 EXPERIMENTS AND NEW VARIATIONAL ARCHITECTURES

Using the well-established Variational Auto-Encoder (VAE) framework on MNIST and CIFAR-10 datasets, we show that the stabilized KS enables reliable training as both a variational posterior [Eqns. (8)–(10)] and likelihood function [Eqns. (4)–(7)]. We then introduce two new deep variational architectures that leverage bounded interval-supported latent variables: the Variational Bandit Encoder (VBE) for improving exploration-exploitation trade-offs in contextual multi-armed bandits (Section 4.2), and the Variational Edge Encoder (VEE) for enhancing uncertainty quantification in link-prediction with graph neural networks (Section 4.3). These novel architectures tend to be most performant when using the KS as their variational posterior. Across the experimental domains, our stable KS tends to be more performant and easier to use than alternative bounded interval-supported variational distributions. For instance, $\tanh_{\mathcal{N}}$ models require log-pdf clipping for stability, while Beta models show significant performance variability based on the chosen positivity-enforcing link function and often fail to converge, e.g., on CIFAR-10 in Section 4.1. Finally, our new variational models are fast: the VBEs in Section 4.2 are $8 - 22\times$ faster than the state-of-the-art baseline.

*Remark* 1. **Across all three experimental settings, models using the unstable KS produce NaN errors in training and are therefore excluded**. Prior work using the KS in latent variable models (Nalisnick et al., 2016; Nalisnick & Smyth, 2017; Stirn et al., 2019) similarly find NaN errors, and rely on parameter clamping $(a_{\min}, a_{\max}), (b_{\min}, b_{\max})$ and uniform base distribution constraints $(u_{\min}, u_{\max})$ to avoid instability. While feasible for small-scale models addressed in such prior work ($\approx 10^2$ KS latent variables), this approach is impractical for the large-scale settings addressed in this work ($10^7$ latent variables), where an instability in a single KS latent can cause training failure. Our stabilization approach directly resolves these numerical issues, eliminating the need for such hyperparameter tuning and enabling stable training at scale. By ensuring robust computation, our method prevents catastrophic failures in large models and simplifies development workflows.

### 4.1 IMAGE VARIATIONAL AUTO-ENCODERS

The VAE (Kingma & Welling, 2014) is a generative latent variable model trained using amortized variational inference. Both the variational posterior $q_{\phi}(\boldsymbol{z}|\boldsymbol{x})$ and the conditional likelihood $p_{\theta}(\boldsymbol{z}|\boldsymbol{x})$ are parameterized using NNs, known as the encoder $e_{\phi}(\boldsymbol{x}) : \mathbb{R}^M \mapsto \mathbb{R}^D$ and decoder $d_{\theta}(\boldsymbol{z}) : \mathbb{R}^D \mapsto \mathbb{R}^M$, respectively. VAEs typically use the standard Normal distribution as the prior and a factorized Normal as the variational posterior. The use of alternative variational distributions allows incorporating different prior assumptions about the latent factors of the data, such as bounded support or periodicity (Figurnov et al., 2018).

**Experimental setup and metrics.** Inspired by (Loaiza-Ganem & Cunningham, 2019), we train VAEs with fully factorized priors and variational posteriors on MNIST and CIFAR-10 without pixel binarization, using an unmodified ELBO ($\beta_{\text{KL}} = 1$). We adopt the most effective likelihoods from their work (Beta and $\mathcal{CB}$), identical latent dimension $D$ (MNIST: $D = 20$, CIFAR-10: $D = 50$), and the same standard NN architectures, which are detailed in Appendix A.5, along with the training

**Algorithm 1** Variational Bandit Encoder

**Require:** $\{\mathbf{x}_k\}_{k=1}^K$, $\{v_k\}_{k=1}^K$, $\eta$, $\beta_{\mathrm{KL}}$
1: Variation posterior $q \leftarrow$ KS
2: Replay buffer $\mathcal{D} \leftarrow \emptyset$
3: **for** $t = 1 \ldots T$ **do**
4:     Encode: $(a_k, b_k) = e_{\boldsymbol{\phi}}(\mathbf{x}_k)$
5:     Sample: $\tilde{z}_k \sim q(z_k; a_k, b_k)$
6:     TS: $a = \operatorname{argmax}_k\{\tilde{z}_k\}$
7:     Reward: $r \sim \mathrm{Ber}(v_a)$
8:     $\mathcal{D} \leftarrow \mathcal{D} \cup \{(\boldsymbol{x}_a, a, r)\}$
9:     Construct $\hat{\mathcal{L}}_{\beta_{\mathrm{KL}}}$ as in (11)
10:     $\boldsymbol{\phi} \leftarrow \boldsymbol{\phi} + \eta \nabla_{\boldsymbol{\phi}} \hat{\mathcal{L}}_{\beta_{\mathrm{KL}}}$
11: **end for**

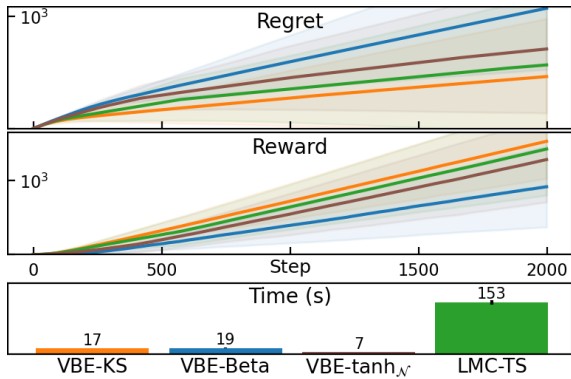

Figure 4: Synthetic bandit performance over 5 runs. VBE-KS best handles explore-exploit trade-offs.

hyperparameters. For each variational posterior factor, we choose the canonical prior: $\mathcal{N}_{(0,1)}$ for $\mathcal{N}$, and $U_{(0,1)}$ for KS and Beta. We evaluate the models using test Log Likelihood (LL), approximated by decoding a single sample from the encoded posterior and computing the log conditional likelihood. To assess usefulness of the learned latent representations, we encode test data $\boldsymbol{x}_n$, compute the mean $\mathbb{E}[q_{\boldsymbol{\phi}}(\boldsymbol{z}_n|\boldsymbol{x}_n)]$, and classify the test labels using a 15-nearest neighbor classifier; the classifier accuracy (%) is denoted $\mathcal{K}(\boldsymbol{\phi})$. For subjective evaluation, we display the mean decoded likelihood of a single sample from the encoded posterior of random test digits in Table 2.

**Discussion of results.** The sole purpose of this experiment is to provide evidence toward the stabilization of the KS. Notably, stable KS VAEs maintain numerical stability while all VAEs with the unstable KS produce unstable training. VAEs with Beta-distributed variational posteriors often do not converge; indeed, (Figurnov et al., 2018) reported strong performance on binarized MNIST using a softplus link function, but did not present results on CIFAR-10, nor could we find other works that did. We suspect this is due to similar instability issues, with the higher gradient variance of the Beta's implicit reparameterization a likely explanation. In an attempt to overcome this instability in Beta VAEs we report the best metrics across softplus or $\exp$ link functions in Table 1. When neither converges, we report N/A. The results in Table 1 show that across datasets, VAEs with KS-distributed variational posteriors consistently produce useful latent spaces, evidenced by strong $\mathcal{K}(\boldsymbol{\phi})$, and yield reconstructions with high LLs and visual quality.

When paired with any variational posterior, a KS likelihood yields stronger MNIST reconstructions than Beta likelihoods: compare rows $*$-Beta to $*$-KS in Table 2. As in (Loaiza-Ganem & Cunningham, 2019), we find $\mathcal{CB}$ likelihoods produce the most subjectively performant VAEs on MNIST, unsurprising as $\mathcal{CB}$ was introduced specifically for the approximately binary MNIST pixel data.

## 4.2 Contextual Bernoulli multi-armed bandits

The contextual Bernoulli multi-armed bandit (MAB) problem involves a decision maker who, at each time step $t = 1, \ldots, T$, selects one arm from a finite set of $K$ options. Each arm has an associated context $\boldsymbol{x}_k \in \mathbb{R}^d$, and pulling an arm yields a binary reward $r_k \sim \mathrm{Ber}(v_k)$, where $v_k \in [0, 1]$ is the unknown mean reward. MABs originate by analogy to casino slot machines, where each machine (arm) has a different payout rate, and the challenge lies in deciding which arms to pull in order to maximize total winnings while learning about their payout rates, a situation called the exploration-exploitation dilemma. MABs have found applications in modern recommendation systems (Li et al., 2010), clinical trials design (Villar et al., 2015), and mobile health (Tewari & Murphy, 2017). Thompson Sampling (TS) is a simple, empirically effective (Chapelle & Li, 2011), and scalable (Jun et al., 2017) arm selection heuristic. It selects the arm corresponding to the highest value drawn from the posterior distributions over the latent $z_k$'s. This approach naturally balances exploration and exploitation: the uncertainty in the posteriors promote exploration, while concentration of probability mass on large mean rewards drive exploitation.

**Variational Bandit Encoder (VBE): VAE $\cap$ TS.** Our novel VBE posits a fully factorized KS variational posterior $\prod_k q_{\boldsymbol{\phi}}(z_k|\boldsymbol{x}_k)$, prior $p(\boldsymbol{z}) = U_{(0,1)}^K$, and a Bernoulli reward likelihood $p(r|v_k)$

for each arm. Similar to VAEs, we employ amortized inference using a shared NN encoder $e_{\phi}(\boldsymbol{x}_k)$, which defines a reparameterizable variational distribution $q_{\phi}(z_k|\boldsymbol{x}_k)$. However, unlike VAEs, VBEs omit the decoder; samples $\tilde{z}_k \sim q_{\phi}(z_k|\boldsymbol{x}_k)$ directly parameterize the reward likelihood. The arm selection at step $t$ follows TS: $a = \arg\max_k\{\tilde{z}_k\}$. We then draw reward $r \sim \text{Ber}(v_a)$ and record it in the replay buffer $\mathcal{D} \leftarrow \mathcal{D} \cup \{(\boldsymbol{x}_a, a, r)\}$. We construct a sample approximation of the modified ELBO over the subset of arms $\mathcal{K}_t \subset \{1, \ldots, K\}$ that have been pulled by time $t$ as

$$\hat{\mathcal{L}}_{\beta_{\text{KL}}}(\mathcal{D}, \boldsymbol{\phi}) = \sum_{(\boldsymbol{x}_a, a, r) \in \mathcal{D}} \log p(r|\tilde{z}_a) + \beta_{\text{KL}} \sum_{k \in \mathcal{K}_t} \mathcal{H}[q_{\phi}(z_k|\boldsymbol{x}_k)], \tag{11}$$

see Appendix A.4 for the derivation. The second term promotes exploration by penalizing overconfidence with the exploration effect proportional to $\beta_{\text{KL}}$. We maximize (11) w.r.t. $\phi$ via gradient ascent, enabled by the reparameterizable KS. VBE execution is summarized in Algorithm 1.

**VBE advantages.** VBEs provide four primary advantages over alternative TS-based Bernoulli MAB approaches, discussed in Section 5.

- *Scalability and Compatibility.* VBE training consists of a forward pass through a NN, sampling an explicitly reparameterized distribution, and a backward pass for gradient-based updates. This process is scalable and fully compatible with existing gradient-based infrastructure.

- *Prior Knowledge Incorporation.* When prior knowledge exists on an arm $k$ it can be efficiently encoded as $p(z_k) = \text{Beta}(a_k, b_k)$, replacing $\mathcal{H}[q_{\phi}(z_k|\boldsymbol{x}_k)]$ with $-D_{\text{KL}}(q_{\phi}(z_k|\boldsymbol{x}_k) \| p(z_k))$.

- *Interpretability and Independence.* Encoding $\boldsymbol{x}_k$ produces KS distribution parameters, fully encapsulating the model's beliefs about $v_k$. This is independent of other arms and past data.

- *Simplicity.* VBEs lack numerous hyperparameters and complex architectural components.

Alternative methods lack some or all of these properties because they do not directly model the mean rewards nor differentiate through mean reward samples; instead, they model the parameters $\phi$.

**Experimental setup.** We construct synthetic data with $K = 10^4$ arms and $T = 2 \cdot 10^3$ steps, sample vector $\boldsymbol{w}$ and features $\{\boldsymbol{x}_k\}_{k=1}^K$ from $\mathcal{N}(\boldsymbol{0}, \boldsymbol{I}_5)$, min-max normalize $\{\boldsymbol{w}^\top \boldsymbol{x}_k\}_{k=1}^K$ to produce probabilities, and further raise them to power 5 to add non-linearity; see Appendix A.6 for details. We evaluate VBEs with either a KS (VBE-KS), Beta (VBE-Beta), or $\tanh_{\mathcal{N}}$ (VBE-$\tanh_{\mathcal{N}}$) all using $\beta_{\text{KL}} = |\mathcal{K}_t|^{-1}$, which makes the second term in (11) a mean. VBE-$\tanh_{\mathcal{N}}$'s performance is sensitive to the number of samples used in its entropy estimate: we found degraded performance beyond 10 samples. The learning rate is set to $\eta = 10^{-2}$. As a baseline, we use LMC-TS, which employs Langevin Monte Carlo (LMC) to sample posterior parameters of a NN, known for state-of-the-art performance across various tasks (Xu et al., 2022). All models use an MLP with 3 hidden layers of width 32. LMC-TS hyperparameters (inverse temperature, LMC steps, weight decay) are set or tuned based on the authors' code. We repeat experiments 5 times on an Apple M2 CPU and report the mean and standard deviation across these runs in Figure 4.

**Metrics and evaluation.** The optimal policy always selects the arm with the highest mean reward $r^*$. Our objective is to minimize regret, defined as the cumulative difference between the expected reward of the chosen action and the optimal action (accessible in the synthetic setting), i.e., $\sum_{t=1}^T (r^* - r_{a_t})$. VBE-KS achieves lower regret and higher cumulative reward than all baselines. VBE-Beta performs significantly worse than VBE-KS and VBE-$\tanh_{\mathcal{N}}$, highlighting the importance of explicit reparameterization. LMC-TS is performant — worse than VBE-KS and better than VBE-$\tanh_{\mathcal{N}}$ — but is 8–22× slower than VBEs: VBEs avoid the computational overhead of LMC.

### 4.3 VARIATIONAL LINK PREDICTION WITH GRAPH NEURAL NETWORKS

Graph Neural Networks (GNNs) have become a powerful tool for learning from graph-structured data, with applications in critical areas like drug discovery (Zhang et al., 2022) and finance (Wang et al., 2022). A key task is link prediction, where the goal is to infer unobserved edges between nodes. However, real-world deployment of graph learning models is often hindered by a lack of reliable uncertainty estimates and limited capacity to incorporate prior knowledge (Wasserman & Mateos, 2024). To address these challenges, we propose a variational approach where the GNN encodes a KS to model the unobserved probabilities of each network link's existence, enabling uncertainty quantification and prior knowledge integration with minimal computational overhead.

In a typical link prediction setup, the GNN has access to the features $\boldsymbol{X} \in \mathbb{R}^{N \times d}$ of all $N$ nodes, but only a subset of positive edges in the training $\mathcal{D}_{tr}$ and validation $\mathcal{D}_{val}$ sets. Edges are labeled as $1$ (present) or $0$ (absent). The GNN generates edge embeddings through message passing and neighborhood aggregation, outputting probabilities $z_{u,v} \in (0, 1)$ that parameterize a Bernoulli likelihood. The seminal work of (Kipf & Welling, 2016) proposed Variational Graph Auto-encoders (VGAEs), which posits a Gaussian variational posterior over the final *node* embeddings. When used for link prediction it samples final node embeddings from the variational posterior and decodes them to produce edge probabilities. In contrast, our approach directly models the probability of an edge using the KS. An advantage of directly modeling edge probabilities is interpretability; deep nodal embeddings are often difficult to interpret, and priors are typically selected for computational tractability rather than their ability to incorporate meaningful prior information. However, the probability of an edge $(u, v)$ existing between two nodes is an interpretable quantity that can often be informed by domain expertise. For example, in gene regulatory networks, epidemiological networks, and social networks experts often have prior knowledge about the likelihood of specific interactions, transmissions, or friendships, respectively, which can be directly incorporated into edge prior $p(z_{(u,v)})$. We believe the limited exploration of variational modeling for edge probabilities is due to the previous lack of an expressive, stable, explicitly reparameterizable bounded-interval distributions.

**Variational Edge Encoder (VEE).** We propose a fully factorized KS variational posterior $\prod_{(u,v) \in \mathcal{D}_{tr}} q_{\boldsymbol{\phi}}(z_{u,v}|\boldsymbol{X}, \mathcal{D}_{tr})$ with a uniform prior $p(\boldsymbol{z}) = U_{(0,1)}^{|\mathcal{D}_{tr}|}$. The GNN encoder $e_{\boldsymbol{\phi}}$ parameterizes a KS distribution for each possible edge $(u, v)$. The remaining structure is highly similar to VBEs: a single sample $\tilde{z}_{u,v} \sim q_{\boldsymbol{\phi}}(z_{u,v}|\boldsymbol{X}, \mathcal{D}_{tr})$ directly parameterizes the Bernoulli likelihood, and we maximize a sample approximation of the modified ELBO

$$\hat{\mathcal{L}}_{\beta_{\mathrm{KL}}}((\boldsymbol{X}, \mathcal{D}_{tr}), \boldsymbol{\phi}) = \sum_{(u,v) \in \mathcal{D}_{tr}} \log p(z_{(u,v)}|\boldsymbol{X}, \mathcal{D}_{tr}) + \beta_{\mathrm{KL}} \sum_{(u,v) \in \mathcal{D}_{tr}} \mathcal{H}[q_{\boldsymbol{\phi}}(z_{(u,v)}|\boldsymbol{X}, \mathcal{D}_{tr})]. \quad (12)$$

From their similarity with VBEs, VEEs inherit the same advantages outlined in Section 4.2.

**Models, metrics, and datasets.** All models use a 2-layer GNN with Graph Convolutional Network (GCN) layers and a hidden/output nodal dimension of 32. In Base-GNN, an MLP decodes the final nodal embeddings into link probabilities. In VEE-KS/Beta/$\tanh_{\mathcal{N}}$ an MLP parameterizes the KS/Beta/$\tanh_{\mathcal{N}}$ variational distributions; all take $\beta_{\mathrm{KL}} = .05|\mathcal{D}_{tr}|^{-1}$. We use 10 samples in $\tanh_{\mathcal{N}}$'s entropy estimate; more did not produce significant performance differences. We train for 300 epochs, with a learning rate of .01, averaging results over 5 runs with different seeds. The posterior predictive distribution over binary links $p(\boldsymbol{A}|\boldsymbol{X}, D_{tr}) = \int p(\boldsymbol{A}|\boldsymbol{Z})q_{\boldsymbol{\phi}}(\boldsymbol{Z}|\boldsymbol{X}, D_{tr})d\boldsymbol{Z}$ is estimated by using a single sample from each KS/Beta distribution, parameterizing each edge Bernoulli distribution with such samples, followed by sampling binary edges. For Base-GNN we directly sample binary edges from the likelihood. Using 30 posterior predictive samples, we compute the edge-wise posterior predictive mean (pred. mean) and standard deviation (pred. stdv.). We report the Pearson correlation $\rho$ between predictive uncertainty (pred. stdv.) and error ($\ell_1$ difference between pred. mean and the true label), as a measure of uncertainty calibration: useful uncertainty estimates should show strong associations between uncertainty and error. Additionally, we compute area under the ROC curve (AUC) using pred. mean as a predictor. Figure 5 shows performance across 3 standard citation networks: Cora, Citeseer, and Pubmed.

**Discussion of results.** On all datasets and all metrics, VEE-KS outperforms or matches the most performant baselines, providing higher predictive accuracy (AUC) and better uncertainty calibration (higher $\rho$). Similar to Section 4.2, we find Beta distributed variational posteriors perform significantly worse than those using KS or

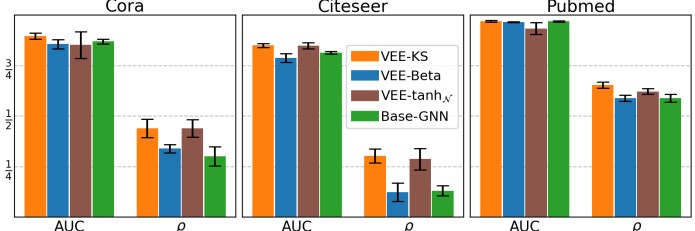

Figure 5: VEE-KS produces informative and calibrated edge posterior predictives across graph datasets.

$\tanh_{\mathcal{N}}$, further underlining the importance of explicit reparameterization. Moreover, models using explicitly reparameterizable latents are faster: on the largest dataset (Pubmed), the average time

(ms) per epoch for VEE-KS, VEE-$\text{tanh}_{\mathcal{N}}$, and VEE-Beta was $381 \pm 61$, $301 \pm 26$, and $447 \pm 86$ respectively, on an Apple M2 CPU.

## 5 RELATED WORK

**VBEs in context: TS-based Bernoulli MAB approaches.** Existing TS-based approaches for Bernoulli MABs assume a prior over model parameters $p(\phi)$, which map contexts to rewards through $e_\phi$. At each round, parameters are sampled from the posterior, $\tilde{\phi}_t \sim p(\phi|\mathcal{D})$, and used to compute mean reward posterior samples $\{e_{\tilde{\phi}_t}(\boldsymbol{x}_k)\}_{k=1}^K$. However, the Bernoulli likelihood often leads to intractable posteriors, making parameter sampling difficult. Common methods use either variational approximations (Chapelle & Li, 2011; Urteaga & Wiggins, 2018; Clavier et al., 2024), primarily Laplace, or MCMC approaches like Gibbs sampling (Dumitrascu et al., 2018) or LMC (Xu et al., 2022). These approaches face several limitations. First, incorporating prior knowledge is challenging since the relationship between a parameter's value and its effect on rewards is often unclear, except in the simplest models. Second, scalability is a concern: Laplace approximations become inefficient with large context dimensions or model sizes, while MCMC-based methods are compute and memory intensive, requiring long burn-in periods (typically $10^2$ iterations) and large machine memory to store the buffer $\mathcal{D}$. Third, interpreting model beliefs over mean rewards requires drawing numerous posterior samples, adding further computational cost. Finally, these methods often introduce significant complexity through intricate algorithms, architectures, optimization steps, and hyperparameters, particularly MCMC parameters (e.g., burn-in iterations, chain length, LMC inverse temperature/weight decay and their respective schedules). By directly modeling mean rewards with a KS, instead of the parameters $\phi$, VBEs offer a simple, scalable, and interpretable approach to Bernoulli MABs.

**Kumaraswamy as a Beta surrogate**. A simple approach to overcome the Beta distribution's lack of explicit reparameterization is to use the KS as a surrogate. This surrogate approach is feasible due to their significant similarities when defined by the same two parameters and the availability of a high-fidelity closed-form approximation of the KL divergence between Beta and KS distributions. (Nalisnick et al., 2016; Nalisnick & Smyth, 2017) use KSs as surrogates for Betas in the Dirichlet Process stick-breaking construction to allow for stochastic latent dimensionality in a VAE. However, both require parameter clipping for numerical stability. In their published code (Nalisnick et al., 2016) constrains KS parameters $\log a, \log b \in [-2.3, 2.9]$, significantly limiting the expressiveness of latent KS distributions. Also, (Nalisnick & Smyth, 2017) comments under a *Computational Issues* section that 'If NaNs are encountered...clipping the parameters of the variational Kumaraswamys usually solve the problem.' (Stirn et al., 2019) improved upon (Nalisnick et al., 2016) by resolving the order-dependence issue in approximating a Beta with a KS. Similarly, (Singh et al., 2017) followed a comparable process using an Indian Buffet Process. Both works maintained numerical stability by restricting the uniform base distribution's support from the unit interval to a narrower interval, before passing the samples through the inverse CDF producing a distortion of the reparameterized sampling distribution. This work eliminates the need for such distortions, enabling more accurate Beta approximations and simplifying the use of the KS distribution by ensuring numerical stability without additional interventions.

## 6 CONCLUSION, LIMITATIONS, AND FUTURE WORK

We identified and resolved key numerical instabilities in the KS distribution, a uniquely attractive option in scalable variational models for bounded latent variables. Our work demonstrates that the stabilized KS can tackle a wide range of large-scale machine learning challenges by powering simple deep variational models. We introduce the Variational Bandit Encoder, which enhances exploration-exploitation trade-offs in contextual Bernoulli MABs, and the Variational Edge Encoder, which improves uncertainty quantification in link prediction using GNNs. Our empirical results show these models are both performant and fast, achieving their best performance with the KS while avoiding the instability and complexity seen in alternatives like the Beta or $\text{tanh}_{\mathcal{N}}$ distributions. These models open avenues for addressing other large-scale challenges, including in recommendation systems, reinforcement learning with continuous bounded action spaces, network analysis, and uncertainty quantification in deep learning, such as modeling per-parameter dropout probabilities using a Concrete distribution (Gal et al., 2017).

KS generalizations (Usman & ul Haq, 2020) inherit $\log(1 - \exp(x))$ instabilities, which future work can resolve by building on our stabilization technique. A limitation of the current models is their inability to capture multimodal posteriors. Future work could explore KS mixtures or hierarchical latent spaces to bridge this gap. Further, optimizing the $\beta_{\mathrm{KL}}$ parameter with techniques like warm-up schedules could yield further performance gains (Alemi et al., 2018). Applications of our stable KS distribution to non-parametric models like the Dirichlet Processes follows directly from prior work (Nalisnick & Smyth, 2017; Stirn et al., 2019). Lastly, a theoretical analysis of the VBE, particularly in proving regret bounds, could extend its applicability to critical areas like clinical trials, where robust decision-making under uncertainty is essential.

## REPRODUCIBILITY STATEMENT

We have made our anonymized code publicly available as supplementary material accompanying this submission. Algorithmic details including hyperparameter selections are given in the body, and included in config files in the code. Additional details regarding data generation for the MAB experiment are included in Appendix A.6.

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

# A APPENDIX

## A.1 BOUNDED INTERVAL-SUPPORTED DISTRIBUTIONS

| Property / Distributions | $\mathcal{CB}$ | $\tanh_{\mathcal{N}}$ | Beta | KS |
|---|---|---|---|---|
| Expressiveness | low | high | high | high |
| Gradient Reparam. | explicit | explicit | implicit | explicit |
| Numerical Issues | mild | high | low | low |
| Complex Functions | $\tanh^{-1}$ | $\log(1\text{-}\tanh^2(x))$ | $\beta, I$ | None |
| Parameterization | $\log \lambda \in \mathbb{R}$ | $\mu, \log \sigma \in \mathbb{R}$ | $a, b \in \mathbb{R}_+$ | $\log a, \log b \in \mathbb{R}$ |
| Analytical Moments | ✓ | ✗ | ✓ | ✓ |
| Closed-form KL Functions | Exp. Family | $\tanh_{\mathcal{N}}$ | Exp. Family | Beta |
| Entropy $\mathcal{H}$ | ✓ | ✗ | ✓ | ✓ |

Table 3: Comparison of bounded interval-supported distribution families.

Table 3 compares workhorse bounded-interval supported distribution families across important properties for latent variable modeling. *Expressiveness* refers to the variety of prototypical shapes each distribution can represent. All but the $\mathcal{CB}$ distribution exhibit four shapes; $\mathcal{CB}$ is limited to two. For more details, see Figure 1 (right) and Section 2. *Numerical issues* highlight the challenges in stable evaluation of some distribution-related function. The $\mathcal{CB}$ requires a Taylor expansion to avoid singularities when its parameter $\lambda$ approaches $0.5$. Similarly, the $\tanh_{\mathcal{N}}$ distribution requires log-pdf clipping and parameter regularization, as appears in various implementations (Haarnoja et al., 2018). *Complex functions* refer to any operation in a distribution-related function that are not affine, logarithmic, or the exponential. The $\tanh_{\mathcal{N}}$ involves computing $\log\left(1 - \tanh^2(x)\right)$, which poses stability challenges (Björck et al., 2021). The Beta distribution requires the Beta function $\beta$ and the regularized incomplete Beta function $I$, both of which rely on numerical approximation. In contrast, the KS distribution in our parameterization avoids complex functions; note $a^{-1}$ is computed via $\exp(-\log a)$, avoiding division. *Parameterization* evaluates whether the distribution can be effectively expressed using unconstrained parameters; all but the Beta have such an ability. *Closed-form KL functions* refer to the availability of closed-form KL divergence expressions with other distributions. All but $\tanh_{\mathcal{N}}$ have such expressions with exponential family members, whose simple moment expressions facilitate easier prior modeling. *Entropy $\mathcal{H}$* refers to the availability of a closed-form expression for differential entropy. This is available for all but the $\tanh_{\mathcal{N}}$ distribution.

## A.2 PRECISION ENHANCING FUNCTIONS

When $|x| \ll 1$, both $\log(1+x)$ and $\exp(x)-1$ suffer from severe cancellation: the former between $1$ and $x$, the latter between $\exp(x)$ and $-1$. In both cases, a simple solution for accurate computation in the presence of small $|x|$ is to use a few terms of the Taylor series, as

$$\texttt{log1p}(x) := \log(1+x) = x - \frac{x^2}{2} + \frac{x^3}{3} - \dots, \quad \text{for } |x| < 1,$$

$$\texttt{expm1}(x) := \exp(x) - 1 = x + \frac{x^2}{2!} + \frac{x^3}{3!} + \dots, \quad \text{for } |x| < 1,$$

where $n!$ denotes the factorial.

## A.3 COUNTER INTUITIVE STABILITY PROPERTIES OF THE UNSTABLE KS

When using the unstable KS to model latent variables with SVI, instability can *worsen with increasing evidence*. Here, SVI will leverage the inverse CDF and its gradient expressions (8)–(10), which depend on the term $w_{b^{-1}}(u) = \log(1 - \exp(b^{-1} \log u))$, to approximate the gradient of the ELBO. Consider modeling the latent probability of heads in coin flipping using a KS, where the true probability is $0.5$. With a uniform prior and few flip observations, the posterior will be well approximated with a low entropy bell-shaped KS, representable with low magnitude parameters $a, b > 1$, keeping $b^{-1}$ away from zero. This avoids catastrophic cancellation in $1 - \exp(b^{-1} \log u)$, as $\exp(b^{-1} \log u)$

remains far from 1. However, as more flips are observed, the posterior sharpens (attains higher entropy), requiring larger values of $b$ to represent the increasing certainty. This pushes $\exp(b^{-1}\log u)$ closer to 1, increasing the risk of catastrophic cancellation and leading to numerical instability. We believe this counter-intuitive behavior likely frustrated modelers, but is no longer an issue in the stabilized KS.

## A.4 VBE MODIFIED ELBO DERIVATION

Let $\mathbf{X} = [\mathbf{x}_1, \ldots, \mathbf{x}_K]$ be a matrix where the $k$-th column corresponds to the context feature $\mathbf{x}_k$. Assuming independence between arms and within-arm rewards, the data likelihood can be factorized as $p(\mathcal{D}|\mathbf{z}) = \prod_{(\mathbf{x}_a, a, r) \in \mathcal{D}} p(r|z_a)$. We adopt a fully factorized variational posterior of the form $q_\phi(\mathbf{z}|\mathbf{X}) = \prod_{k=1}^K q_\phi(z_k|\mathbf{x}_k)$. Recall that $\mathcal{K}_t \subset \{1, \ldots, K\}$ represents the subset of arms that have been pulled, and thus for which we have reward data.

The modified ELBO is derived as follows:

$$
\begin{aligned}
\mathcal{L}_\beta(\mathcal{D}, \phi) &= \mathbb{E}_{q_\phi(\mathbf{z}|\mathbf{X})}[\log p(\mathcal{D}|\mathbf{z})] - \beta_{\mathrm{KL}}\mathrm{KL}\left(q_\phi(\mathbf{z}|\mathbf{X}) \,\|\, p(\mathbf{z})\right) \\
&= \mathbb{E}_{q_\phi(\mathbf{z}|\mathbf{X})}[\log p(\mathcal{D}|\mathbf{z})] + \beta_{\mathrm{KL}}\mathcal{H}\left[q_\phi(\mathbf{z}|\mathbf{X})\right], \quad p(\mathbf{z}) = U_{(0,1)}^K \\
&= \mathbb{E}_{q_\phi(\mathbf{z}|\mathbf{X})}[\log p(\mathcal{D}|\mathbf{z})] + \beta_{\mathrm{KL}} \sum_{k \in \mathcal{K}_t} \mathcal{H}\left[q_\phi(z_a|\mathbf{x}_a)\right] \\
&= \mathbb{E}_{q_\phi(\mathbf{z}|\mathbf{X})}\left[\sum_{(\mathbf{x}_a, a, r) \in \mathcal{D}} \log p(r|z_a)\right] + \beta_{\mathrm{KL}} \sum_{k \in \mathcal{K}_t} \mathcal{H}\left[q_\phi(z_a|\mathbf{x}_a)\right] \\
&\approx \sum_{(\mathbf{x}_a, a, r) \in \mathcal{D}} \log p(r|\tilde{z}_a) + \beta_{\mathrm{KL}} \sum_{k \in \mathcal{K}_t} \mathcal{H}\left[q_\phi(z_a|\mathbf{x}_a)\right], \quad \tilde{z}_a \sim q_\phi(z_a|\mathbf{x}_a)
\end{aligned}
$$

where in the final step, we use a single sample approximation of the expectation.

## A.5 VAE ARCHITECTURAL AND TRAINING CHOICES

The following is almost identical to that used in (Loaiza-Ganem & Cunningham, 2019), but provided here for completeness. For both experiments (MNIST and CIFAR-10) we use a learning rate of 0.001, batch size of 500, and optimize with Adam for 200 epochs.

**Enforcing positive variational parameters.**

- *Gaussian.* When the variational posterior is Normal, the output layer of the encoder uses a softplus nonlinearity for the positive standard deviation.

- *KS.* As we parameterize the KS by unconstrained $\log$ values, any required exponentiation occurs internally, so we require no nonlinearity on the output of the encoder.

- *Beta.* The core software libraries do not implement the Beta distribution's reparameterized sampling with unconstrained $\log$ parameter values, so we use an exponential nonlinearity on the output of the encoder to enforce positivity. A softplus nonlinearity was attempted which was found to be less stable likely due to the model seeing very large latent parameter values, which is more stably accessible via an $\exp$.

**Enforcing positive likelihood parameters.**

- *$\mathcal{CB}$.* When the likelihood is a $\mathcal{CB}$, the output of the decoder has a sigmoid non-linearity to enforce its parameter $\lambda \in (0,1)$.

- *KS.* As we parameterize the KS by unconstrained $\log$ values, any required exponentiation occurs internally, so we require no further transformation on the output of the decoder.

- *Beta.* The core software libraries do not implement the Beta distribution's log-pdf with unconstrained $\log$ parameter values, so we use a softplus nonlinearity on the output of the decoder to enforce positivity. An exponential nonlinearity was attempted which was found to be less stable.

Table 4: VAE on MNIST and CIFAR-10 with standard deviation.

| Prior | $q_\phi(z|x)$ | $p_\theta(x|z)$ | MNIST | | CIFAR-10 | |
|---|---|---|---|---|---|---|
| | | | ELBO | $\mathcal{K}(\phi)$ | ELBO | $\mathcal{K}(\phi)$ |
| $\mathcal{N}_{(0,1)}$ | $\mathcal{N}$ | $\mathcal{CB}$ | $1825 \pm 98$ | 97.3 | $1167 \pm 901$ | 37.9 |
| $U_{(0,1)}$ | KS | $\mathcal{CB}$ | $1818 \pm 104$ | 97.4 | $1172 \pm 908$ | 41.5 |
| $U_{(0,1)}$ | Beta | $\mathcal{CB}$ | $1821 \pm 98$ | 97.5 | $1167 \pm 907$ | 40.3 |
| $\mathcal{N}_{(0,1)}$ | $\mathcal{N}$ | Beta | $4073 \pm 5701$ | 92.1 | $3566 \pm 1203$ | 48.5 |
| $U_{(0,1)}$ | KS | Beta | $4061 \pm 1932$ | 91.3 | $3483 \pm 1133$ | 50.1 |
| $U_{(0,1)}$ | Beta | Beta | $4082 \pm 1522$ | 90.1 | N/A | N/A |
| $\mathcal{N}_{(0,1)}$ | $\mathcal{N}$ | KS | $3328 \pm 989$ | 96.4 | $1720 \pm 884$ | 47.1 |
| $U_{(0,1)}$ | KS | KS | $3355 \pm 512$ | 96.8 | $1738 \pm 877$ | 48.8 |
| $U_{(0,1)}$ | Beta | KS | $3348 \pm 515$ | 97.1 | N/A | N/A |

**Data augmentation for** $(0,1)$ **likelihood functions.** The $\mathcal{CB}$ has support $[0,1]$ and handles data on the support boundaries without issue. When the likelihood function is a Beta or KS, which have support $(0,1)$, we clamp pixel intensities to $\left[\frac{1}{2\times255}, 1 - \frac{1}{2\times255}\right]$ to prevent non-finite gradient values.

For all our MNIST experiments we use a latent dimension of $D = 20$, an encoder with two hidden layers with 500 units each, with leaky-ReLU non-linearities, followed by a dropout layer (with parameter 0.9). The decoder also has two hidden layers with 500 units, leaky-ReLU non-linearities and dropout. For all our CIFAR-10 experiments we use a latent dimension of $D = 40$, an encoder with four convolutional layers, followed by two fully connected ones. The convolutions have respectively, 3, 32, 32 and 32 features, kernel size 2, 2, 3 and 3, strides 1, 2, 1, 1 and are followed by leaky-ReLU non-linearities. The fully connected hidden layer has 128 units and a leaky-ReLU non linearity. The decoder has an analogous "reversed" architecture.

**Expanded Experimental Results**. Table 4 includes identical data from image VAE experiments from Section 4.1, but now with the standard deviations across test samples included for the ELBO.

A.6 BERNOULLI MULTI-ARMED BANDIT DATA GENERATION

In Section 4.2, we generate synthetic data for $K = 10^4$ arms by first sampling a weight vector $\mathbf{w}$ and features $\{\mathbf{x}_k\}_{k=1}^K$ from $\mathcal{N}(\mathbf{0}, \mathbf{I}_5)$. We then compute $\{\mathbf{w}^\top \mathbf{x}_k\}_{k=1}^K$ and apply min-max normalization to produce probabilities (referred to as "Original probabilities" in Figure 6). To introduce non-linearity, we raise these probabilities to the power 5 (shown as "Power (5) transformed probabilities" in Figure 6).

Exponentiating the probabilities not only makes the mapping from features to mean rewards more challenging to learn, but it also significantly reduces the number of arms with high probabilities, forcing the agent to explore more. For instance, when raising the probabilities to the power of 5, the number of arms with large probabilities drops from 167 to just 7.

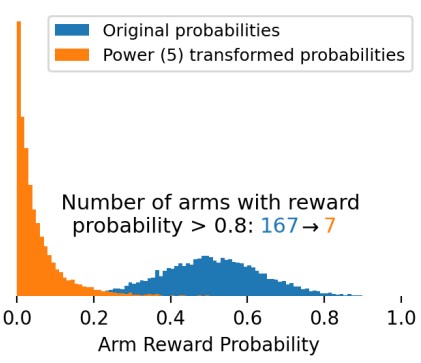

Figure 6: High arm reward probabilities are reduced via a power 5 exponentiation, encouraging exploration.

A.7 KUMARASWAMY DIFFERENTIAL ENTROPY

For a continuous distribution $q$ with interval support $(l, h)$, the differential entropy $\mathcal{H}(q)$ is equal to the KL divergence between $q$ and $U_{(l,h)}$ plus a constant proportional to the interval's width $h - l$:

$$D_{\text{KL}}\left(q \,\|\, U_{(l,h)}\right) := -\mathbb{E}_q\left[\log\frac{q}{U_{(l,h)}}\right] = -\mathbb{E}_q\left[\log q\right] + \mathbb{E}_q\left[\log U_{(l,h)}\right] = \mathcal{H}(q) + \log\frac{1}{h-l}.$$

When the support has $h - l = 1$, then $D_{\text{KL}}\left(q \,\|\, U_{(0,1)}\right) = \mathcal{H}(q)$. Then the differential entropy of a KS with parameters $a, b$ is

$$\mathcal{H}(\text{KS}) = D_{\text{KL}}\left(\text{KS} \,\|\, U_{(0,1)}\right) = 1 - b + (1 - a)\left(\phi^{(0)}\left(b^{-1} + 1\right) + \gamma\right) - \log a - \log b,$$

where $\phi^{(0)}$ is the digamma function and $\gamma \approx 0.577$ is the Euler-Mascheroni constant. The digamma function and its gradient, the trigamma function $\phi^{(1)}(x)$, can represented as infinite series which converge rapidly and thus can be used effectively in numerical applications. They are included as standard functions in common auto-differentiation frameworks.

## A.8 KUMARASWAMY-BETA KL DIVERGENCE

The KL divergence between the Kumaraswamy distribution $q(v)$ with parameters $a, b$ and the Beta distribution $p(v)$ with parameters $\alpha, \beta$ is given by:

$$\mathbb{E}_q\left[\log\frac{q(v)}{p(v)}\right] = \frac{a - \alpha}{a}\left(-\gamma - \Psi(b) - \frac{1}{b}\right) + \log ab + \log\mathcal{B}(\alpha, \beta) - \frac{b - 1}{b}$$
$$+ (\beta - 1)b \sum_{m=1}^{\infty} \frac{1}{m + ab}\mathcal{B}\left(\frac{m}{a}, b\right)$$

where $\gamma$ is Euler's constant, $\Psi(\cdot)$ is the Digamma function, and $\mathcal{B}(\cdot)$ is the Beta function. The infinite sum in the KL divergence arises from the Taylor expansion required to represent $\mathbb{E}_q[\log(1 - v_k)]$; it is generally well approximated by the first few terms.

