# OpenReview forum: "Stabilizing the Kumaraswamy Distribution"
_ICLR.cc/2025/Conference — ICLR 2025 Conference Withdrawn Submission_

### Official Review · Reviewer_e7j1 · 2024-10-28

**Soundness:** 3
**Presentation:** 3
**Contribution:** 2
**Rating:** 5
**Confidence:** 3

**Summary:**

This paper introduces modifications to the Kumaraswamy distribution to improve its numerical instability.

**Strengths:**

The analysis of identifying the instability is interesting enough. Also, the modification of the distribution makes sense a lot.

**Weaknesses:**

- However, given that large-scale problems are more related with their network capacity and the data, rather than the distribution modeling itself, I doubt if this modification could be benefit to the large-scale training. For example, let's focus on Section 4.1 with Image VAE. Could the authors provide additional argument why do we need this algorithm given that the diffusion model is all based on the normal distribution? I'm more like, why do we need this algorithm given that there is already some well-developed algorithm with another approach? Can we compete with diffusion models with this distribution? I understand different approaches could perform differently, and I'm not completely opposing different approaches. But what I don't see is the practicality of this algorithm.

**Questions:**

-

**Details Of Ethics Concerns:**

-

---

> ### Author Response · Authors · 2024-11-21
> **Authors’ response to Reviewer e7j1**
>
> Thank you for taking the time to review our manuscript and for recognizing our work as interesting and well-motivated. Below, we address your concerns regarding the practicality of our algorithms and highlight the significant novelty and contributions which were perhaps overlooked in your initial assessment.
>
>
> &nbsp;
>
> **Practicality of a Kumaraswamy VAE**.
>
> The image VAE experiment in Section 4.1 is designed solely as a proof-of-concept to demonstrate the stability of the stabilized KS distribution in a familiar experimental framework. We do not advocate use of the KS in such settings, where other approaches (e.g., diffusion models) are preferred for their specific strengths.
>
> To clarify this, we have revised the opening of Section 4.1’s Discussion of Results to explicitly state: "The sole purpose of this experiment is to provide evidence toward the stabilization of the KS."
>
> &nbsp;
>
> **Four Primary Contributions**
>
> We would like to bring to your attention the four main contributions of this work which may have been overlooked in the initial assessment; these contributions are stated in the introduction and we provide significant evidence for them in the experiments.
>
> &nbsp;
>
> *Two Kumaraswamy (KS) Related Contributions*.
> - Reparameterization of KS functions: Equations 4–10 reparameterize the distribution-functions in terms of unconstrained logarithmic parameters. Their algebraic form makes the dominant $\log(1-\exp(x))$ instabilities explicit and isolated.
> - Stabilization via $\texttt{log1mexp}$: Isolating the the dominant $\log(1-\exp(x))$ instabilities allows for substitution with stable $\texttt{log1mexp}$
>
> The first contribution is a prerequisite to the stabilizing application of $\texttt{log1mexp}$. It further enhances compatibility with neural network settings by removing the need for positivity-enforcing link functions and allowing most computation to stay in log-space.
>
> We hope you share the perspective of other reviewers regarding the significant potential impact *and practicality* of our KS-related contributions, despite their apparent simplicity. Reviewer 3tyx acknowledges these contributions as “quite simple [and] I do think it has important impact in the domain,” while Reviewer xx8r highlights their practicality, noting that “the primitives (expm1() and log1p()) are already present in various frameworks. **This has the potential of rapid adoption in practical settings**.”
>
> &nbsp;
>
> *Two New Latent Variable Models*.
>
> These architectures are NOT merely applications of the KS, but in fact completely new large-scale bounded-latent variable models. They only relate to the KS in that they achieve their best performance when combined with our stabilized KS distribution. This further motivates the attractiveness of the KS in latent variable modeling.
>
> - Variational Bandit Encoders (VBE) address contextual multi-armed bandits, achieving state-of-the-art performance (outperforming LMC-TS) in a fraction of the run-time, while being simple, scalable, and capable of incorporating prior knowledge.  It represents the first such contextual bandit approach to perform direct variational modeling of the expected reward posterior, and leverages the reparameterization trick for efficient large-scale training. We expect this to have broad impact due to the importance of multi-armed bandits in, e.g. recommendation systems, and clinical trials.
> - Variational Edge Encoders (VEE) are a new method for link prediction in GNNs, leveraging bounded latent variables for improved uncertainty quantification.
>
>
> &nbsp;
>
> We suspect the recognized contribution of your initial assessment may be limited to the second ($\texttt{log1mexp}$ substitution) KS related contribution. To highlight the additional novelty of the VBE and VEE, in the revised manuscript we have renamed Section 4 to *“Experiments and New Variational Architectures.”*
>
> &nbsp;
>
> **Closing Remarks**
>
> We thank you for your thoughtful review and for recognizing our work as interesting and well-motivated. We hope our response clarifies the full scope of our contributions - a fundamental reparameterization enabling stable KS distribution and two novel architectures with significant impact in real-world applications.
>
> *Having addressed your single concern regarding practicality, we kindly ask you to consider revising your score to reflect the work’s significantly broader contributions and impact, which may not have been fully recognized in the initial assessment*.
>
> We strive to improve our paper and will be happy to continue the discussion if any outstanding issues remain.

---

### Official Review · Reviewer_xx8R · 2024-10-30

**Soundness:** 3
**Presentation:** 3
**Contribution:** 2
**Rating:** 5
**Confidence:** 4

**Summary:**

This work identifies and discuss a way to fix numerical instabilities that appear when evaluating the (gradient of the) density and quantile function of the Kumaraswamy (KS) distribution. The authors demonstrate that the computation of $\log(1 - \exp(x))$ is the one that suffers from numerical instabilities due to catastrophic cancellations when $x \rightarrow 0$ and thus $\exp(x) \approx 1$ and underflows. They then employ the numerical approximations from Mächler, which rely on Taylor expansions, to bypass these difficulties and express all computations of the (gradient of the) KS density and quantile function in terms of those. In this way, the authors argue that the KS distribution is stabilized which then allows it to be used as a modelling choice in various settings: image modelling with VAEs, contextual multi-armed bandits and variational link prediction in graph neural networks.

**Strengths:**

* The stabilisation trick is very simple and the primitives (`expm1()` and `log1p()`) are already present in various frameworks. This has the potential of rapid adoption in practical settings.
* Apart from some specific parts on the VBE section, the paper is well presented and written. The authors clearly demonstrate the numerical instabilities, how they manifest and how the proposed modifications can avoid them.
* The tasks that the authors use to evaluate the behaviour of the KS distribution against other alternatives are quite diverse.

**Weaknesses:**

* The novelty of this work is a bit limited. As far as the stabilisation is concerned, while the idea of numerically stable computation is important, the numerical stability is ensured by techniques proposed in the prior work of Mächler. Therefore, in my opinion, the main new insight and novelty of this work is in the benchmarking and the various tasks that the authors considered to test the viability of the KS distribution against other baselines.

* Given the aforementioned, I will thus put more weight on the experimental evaluation. This brings me to the second key weakness; while the authors do consider various other methods for modelling bounded RVs, they are missing the critical baseline of the binary concrete distribution [1] (which is a bit odd, given that they mention in passing the concrete dropout method at the conclusion section). If one treats both the temperature and the  $\alpha$ as free parameters, the binary concrete distribution can have similar representation capacity as the Beta and KS distributions (shown at Figure 1). Furthermore, the authors of [1] also discuss potential numerical instabilities when evaluating the density of the binary concrete and also present in the appendix ways to avoid them via reparametrization (i.e., considering the logistic noise as the random variable and then treating the sigmoid as a downstream invertible transformation). Therefore, this method would also satisfy the criteria that the authors argued for, i.e., supporting the reparametrization trick, provide sufficient expressiveness and offer simple distribution-related functions that are fast and accurate to evaluate.

[1] The Concrete Distribution: A Continuous Relaxation of Discrete Random Variables, Maddison et al., 2017

**Questions:**

Given the strengths and weaknesses mentioned above, I am leaning on the negative side for this work. Some specific questions and remarks that the authors could work on to improve the work are

* The main remark is to include the concrete distribution along with the suggestions for numerical stability discussed at [1] in the evaluation of all tasks considered. This would provide a good amount of information and would allow one to better determine whether the KS distribution is significant for modelling bounded RVs.
* While the authors do show that the KS distribution can be unstable for specific values of the input, they do not really show whether these happen naturally in the tasks considered. Do distribution parameters such as the ones shown in Figure 3 appear in the tasks considered later in the paper? If yes, how does the simple fix from Nalisnick et al. 2016 (i.e., clipping the KS parameters) perform in terms of downstream performance on the tasks considered? This would better highlight the true performance delta of better handling the numerical difficulties.
* For the VBE section, the notation can be improved. There is a $\tilde{z}$, that is for the random sample from the variational posterior over the Bernoulli parameters, and a $z$ that is used for both the ground truth Bernoulli parameter of the bandit arm and the latent variable governed by the variational posterior. I would suggest that the authors separate the last two (e.g., use $q(z_k|x_k)$ for the variational posterior but then use something like $v_a$ for the ground truth Bernoulli parameter) as otherwise the $a = argma{x}_k \tilde{z}_k$, $r\sim Ber(z_a)$ can be confused as taking the payoff from the variational posterior and not the ground truth model.
* For the results at Table 1 I would suggest to the authors to report either the ELBO or approximate log-likelihoods computed by importance sampling (with a sufficient number of samples) and the variational posterior as a proposal. Using a single sample to compute the test LL can be noisy (and there are no error bars at Table 1).
* At line 400 the authors mention the “posterior predictive” as $p(A) = \int p(A|Z)p(Z)dZ$ which is not the posterior predictive but rather the prior predictive / marginal likelihood. The posterior predictive should be $p(A|X, D_tr) = \int p(A|Z)q(Z|X,D_tr)$.

---

> ### Author Response · Authors · 2024-11-21
> **Authors’ response to Reviewer xx8r - Part 1**
>
> Thank you for your thoughtful review and for recognizing the potential for rapid adoption of our stabilized Kumaraswamy (KS) techniques and the clarity of our presentation. Since posting this work, we have observed its adoption in applications we did not initially anticipate, such as compression in vector search, and it is in the process of being integrated into major auto-differentiation libraries.
>
> This response is divided into two parts. First, we clarify the scope of our contributions, which we believe were not fully recognized in the initial assessment. We hope this will shift your focus from benchmarking details to the broader novelty and impact of our work. Second, we provide point-by-point responses to your specific questions and suggestions.
>
>
> &nbsp;
>
> **Four Primary Contributions**
>
> We would like to highlight the four main contributions of this work, which may have been overlooked in your initial assessment. These contributions are detailed in the introduction and supported by our experiments.
>
> &nbsp;
>
> *Two Kumaraswamy (KS) Related Contributions*.
> - Reparameterization of KS functions: Equations 4–10 reparameterize the distribution-functions in terms of unconstrained logarithmic parameters. Their algebraic form makes the dominant $\log(1-\exp(x))$ instabilities explicit and isolated.
> - Stabilization via $\texttt{log1mexp}$: Isolating the the dominant $\log(1-\exp(x))$ instabilities allows for substitution with Mächler’s stable $\texttt{log1mexp}$
>
> The first contribution is a prerequisite to the stabilizing application of $\texttt{log1mexp}$. It further enhances compatibility with neural network settings by removing the need for positivity-enforcing link functions and allowing most computation to stay in log-space.
>
> &nbsp;
>
> *Two New Latent Variable Models*.
>
> These architectures are NOT merely applications of the KS, but in fact completely new large-scale bounded-latent variable models. They only relate to the KS in that they achieve their best performance when combined with our stabilized KS distribution. This further motivates the attractiveness of the KS in latent variable modeling.
>
> - Variational Bandit Encoders (VBE) address contextual multi-armed bandits, achieving state-of-the-art performance (outperforming LMC-TS) in a fraction of the run-time, while being simple, scalable, and capable of incorporating prior knowledge.  It represents the first such contextual bandit approach to perform direct variational modeling of the expected reward posterior, and leverages the reparameterization trick for efficient large-scale training. We expect this to have broad impact due to the importance of multi-armed bandits in, e.g. recommendation systems, and clinical trials.
> - Variational Edge Encoders (VEE) are a new method for link prediction in GNNs, leveraging bounded latent variables for improved uncertainty quantification.
>
> &nbsp;
>
> We suspect the recognized contribution of your initial assessment may be limited to the second ($\texttt{log1mexp}$ substitution) KS related contribution. To highlight the additional novelty of the VBE and VEE, in the revised manuscript we have renamed Section 4 to *“Experiments and New Variational Architectures.”*
>
> &nbsp;
>
> **Refocusing on Novelty and Impact**.
>
> You noted in W1 that the perceived lack of novelty shifted your focus to benchmarking details. We hope the above discussion redirects your attention to the broader novelty and impact of the VBE and VEE, the expanded KS-related contributions (beyond the simple substitution of Mächler’s stable $\texttt{log1mexp}$), and the performance of the KS as a variational posterior across these widely varying tasks. With that said, we address your benchmarking-related concerns point-by-point below.

---

> ### Author Response · Authors · 2024-11-21
> **Authors’ response to Reviewer xx8r - Part 2**
>
> &nbsp;
>
> **Part 2: Point-by-Point Responses; Benchmarking/Experimental Details**
>
> &nbsp;
>
> *Q1: Comparison with Binary Concrete Distribution*.
>
> We appreciate your suggestion to evaluate against the Binary Concrete distribution. We’ve never seen the Concrete distribution used in this manner. Due to the difficulty in its implementation and the need to alter expressions outside the distribution itself to maintain stability in variational settings (e.g. see the altered relaxed ELBO in Equation 27, Appendix C.3.2 of Maddison et al. (2017)), we believe the Binary Concrete distribution does not meet criterion (iii): providing simple distribution-related functions that are fast and accurate to evaluate.
>
> That said, we are eager to provide additional evidence if you feel this remains a critical point despite our attempt to refocus on broader contributions beyond benchmarking. We attempted to implement the Binary Concrete distribution from Maddison et al. (2017) following the details in Appendix Sections C.3.2 and F. Additionally, we tested the implementations provided in PyTorch Distributions and TensorFlow Probability. However, all attempts resulted in failed training. This could be due to the high numerical demands of our large-scale settings (e.g., $10^7$ latent variables), where even a single unstable computation can produce training failure.
>
>
> If you can point us to a preferred reference implementation (preferably in PyTorch), we are happy to test it further.
>
>
> &nbsp;
>
>
> *Q2 (A): Natural Occurrence of KS Instabilities.*
>
> The unstable KS consistently results in training failures (NaN errors) across all experimental settings (VAEs, contextual multi-armed bandits, and link prediction). This provides evidence for the natural occurrence of KS distributions with unstable computation in their distribution-related functions. As noted in Section 4 (lines 239–240): "...models using the unstable KS produce NaN errors and are therefore excluded." To avoid clutter, these failures were omitted from figures, which may have understated their significance.  In the revised manuscript we have made this statement bold to make this point clear.
>
>
> *Q2 (B): Can Clipping Parameters Resolve Instabilities? Clarifying the Impact of such clipping on Experimental Outcomes.*
>
>
> The KS stability issues are not unique to our implementation but reflect a longstanding challenge with the KS distribution, as evidenced by public forum discussions ([1]–[3]) and prior research ([4]–[7]). Prior work ([4]–[7]), including Nalisnick et al. (2016), rely on parameter clamping $(a_{\text{min}}, a_{\text{max}}),  (b_{\text{min}}, b_{\text{max}})$ and uniform base distribution constraints $(u_{\text{min}}, u_{\text{max}})$ to avoid instability. While feasible for small-scale models ($\approx 10^1−10^2$ KS latent variables, as in [4]–[7]), *this approach is impractical for the large-scale settings addressed in this work ($10^7$ latent variables), where an instability in a single KS latent can cause training failure.* In fact, this project was motivated by the failure of such clipping-based approaches; Nalisnick, et. al. (2016)'s parameter range resulted in failed training in our settings.
>
>
> Our stabilization approach directly resolves these numerical issues, eliminating the need for such hyperparameter tuning and enabling stable training at scale. By ensuring robust computation, our method prevents catastrophic failures in large models and simplifies development workflows. This innovation has driven the ongoing integration of our stabilized KS implementation into major auto-differentiation libraries.
>
>
> To clarify these points, we have added *Remark 1* in Section 4 of the revised manuscript.
>
> &nbsp;
>
> *Q3: Improved VBE section notation*.
>
> We agree with your suggestion, which significantly clarifies the VBE section. The revised manuscript incorporates your proposed notation changes.
>
> &nbsp;
>
> *Q4: Refined Evaluation Metrics in Table 1.*
>
> As suggested, we have replaced the test log-likelihood (LL) values in Table 1 with ELBO values. Additionally, we have included an Expanded Experimental Results section in Appendix A.4, with Table 4 reporting ELBO standard deviations for this experiment. The purpose of the VAE experiment is to showcase the stability of the KS distribution in a familiar setting, rather than to advocate for its use in such a setting.
>
> &nbsp;
>
> *Q5: Posterior Predictive Misstatement*.
>
> Thanks for pointing this out. This has been corrected in the updated manuscript.

---

> ### Author Response · Authors · 2024-11-21
> **Authors’ response to Reviewer xx8r - Part 3**
>
> **Closing Remarks**
>
> We appreciate your thoughtful review and agree that the simplicity of this work is a strength, enabling impactful contributions to bounded latent variable modeling and potential for rapid adoption. We further appreciate your recognition of the work's clear exposition and our effort in evaluation across broadly varying tasks. We hope we have clarified the full scope of our contributions — a fundamental reparameterization enabling a stable KS distribution and two novel architectures with significant impact in real-world applications — and refocused your attention on this greatly expanded novelty and broader impact.
>
>
> *We kindly ask you to consider revising your score to reflect the work’s significantly broader contributions and impact, which may not have been fully recognized in the initial assessment*.
>
>
> We strive to improve our paper and will be happy to continue the discussion if any outstanding issues remain.
>
>
> &nbsp;
>
>
> [1] https://github.com/probtorch/pytorch/pull/143#issuecomment-428211244
>
>
> [2] https://github.com/probtorch/pytorch/pull/143#issuecomment-377108639
>
>
> [3] https://github.com/pytorch/pytorch/issues/11937#issuecomment-423587839
>
>
> [4] Eric Nalisnick, Lars Hertel, and Padhraic Smyth. Approximate inference for deep latent Gaussian mixtures. In NeurIPS Worksh. on Bayes. Deep Learn., 2016.
>
>
> [5] Eric Nalisnick and Padhraic Smyth. Stick-breaking variational autoencoders. In ICLR, 2017.
>
>
> [6] Andrew Stirn, Tony Jebara, and David Knowles. A new distribution on the simplex with auto-encoding applications. In NeurIPS, 2019
>
>
> [7] Rachit Singh, Jeffrey Ling, and Finale Doshi-Velez. Structured variational autoencoders for the Beta-Bernoulli process. In NeurIPS Workshop Adv. Approx. Bayes. Infer., 2017.

---

> > ### Comment · Reviewer_xx8R · 2024-11-27
> > **Response to rebuttal**
> >
> > I would like to thank the authors for their response which addressed some of my comments. The revised statement makes clear that the new latent variable models are also a contribution of this work and I will therefore increase my score to a 5. My main remaining concerns are:
> >
> > - The (lack of) comparison against the "simple" fix of clamping (as that would better highlight the improvement against the prior way of stabilising the KS distribution, besides the benefit of eliminating a hyperparameter). The authors do mention that the ranges from prior work did not work, but how difficult is to find an appropriate range? What makes the clamping approach impractical for high dimensional spaces? The constraints are applied elementwise and thus are trivial to implement and enforce, no matter the amount of latent variables. Therefore, finding, e.g, a single $\min$, $\max$ constraint that leads to numerical stability should be straightforward.
> >
> > - If the architectures are also contributions from this work, then more experiments are needed to highlight their importance. For example, why not try the VBE in a recommender system setting (as the authors suggest in the rebuttal) to verify its importance as a specific architecture (independent of the KS modeling assumption)? A similar comment can be made for the VEE; the authors should compare with, e.g., the VGAE as that would better highlight the importance of the modeling assumptions of VEE (again, independent of the KS assumption).

---

### Official Review · Reviewer_275N · 2024-11-06

**Soundness:** 2
**Presentation:** 1
**Contribution:** 1
**Rating:** 3
**Confidence:** 5

**Summary:**

- This work proposes a numerical computation method to stabilize the inverse CDF and log-pdf of Kumaraswamy distribution in deep learning libraries.
- Due to the numerical instability of $\log (1 - \exp (x))$ term, the previously implemented Kumarsawmy distribution has inaccurate pdf and inverse CDF, which is leading to producing corrupted reparameterized samples.
- The authors combine two functions, log(-expm1(x)) and log1p(-exp(x)), to re-formulate $\log (1 - \exp (x))$ term and modified the log-pdf and inverse CDF of the Kumaraswamy.
- Several experiments are conducted to verify the usage of the modified Kumaraswamy.

**Strengths:**

Due to capturing stability during numerical computation, the Kumarswamy can be better utilized in various tasks in the deep learning community.

**Weaknesses:**

While the proposed method enlarges the utilization of the Kumaraswamy, the impact and novelty of the work are limited to a single distribution.

The modified Kumaraswamy should be properly compared with the current Kumaraswamy in the experiments.

**Questions:**

Is there any way to quantitatively compare the modified Kumaraswamy against the previous Kumaraswamy? Such empirical evidence would be required for the comparison.

I guess this can extend some previous works [1,2], as the Beta can be extended to the Dirichlet, right? The extension will definitely strengthen the work.

[1] Stirn et al., A New Distribution on the Simplex with Auto-Encoding Applications, 2019.

[2] Joo et al., Dirichlet Variational Autoencoder, 2020.

---

> ### Author Response · Authors · 2024-11-21
> **Authors’ response to Reviewer 275N - Part 1**
>
> Thank you for your thoughtful review and for recognizing the potential impact of our work on deep latent variable models. Below, we address your comments by discussing the broader significance of the stabilized Kumaraswamy (KS), reviewing the extensive novelty in our contributions (perhaps overlooked in the initial assessment), and addressing the question of quantitatively comparing the stabilized KS to the unstable version.
>
> &nbsp;
>
> **Significance of the stabilized KS and Extensions of Previous Work**
>
> The stabilized KS distribution is a critical tool for latent variable models with bounded variables, meeting the three essential criteria outlined in Section 1: (i) support for the reparameterization trick, (ii) sufficient expressiveness for complex latent spaces, and (iii) offering simple distribution-related functions.
>
> In Section 4, we provide empirical evidence of the stabilized KS’s utility across diverse latent variable models. Beyond these immediate applications, the KS also serves as a building block for constructing more complex distributions and random processes. For example, it can be extended to Dirichlet distributions and nonparametric models, as demonstrated in the works of Stirn et al. (2019) and Joo et al. (2020) which you bring up. Additional uses of the KS are discussed in Section 5 (Related Work: Kumaraswamy as a Beta surrogate).
>
> *The broader impact of the KS extends far beyond the scope of a single distribution*. Its stabilization unlocks potential for a wide range of applications wherever the KS distribution is employed. To make this broader significant clear, we have added explicit commentary in Section 6 (Conclusions, Limitations, and Future Work), highlighting future extensions to e.g. Dirichlet processes and other nonparametric models.
>
> &nbsp;
>
> **Four Primary Contributions**
>
>
> We would like to bring to your attention the four main contributions of this work which may have been overlooked in the initial assessment; these contributions are stated in the introduction and we provide significant evidence for them in the experiments.
>
>
> &nbsp;
>
>
> *Two Kumaraswamy (KS) Related Contributions*.
> - Reparameterization of KS functions: Equations 4–10 reparameterize the distribution-functions in terms of unconstrained logarithmic parameters. Their algebraic form makes the dominant $\log(1-\exp(x))$ instabilities explicit and isolated.
> - Stabilization via $\texttt{log1mexp}$: Isolating the the dominant $\log(1-\exp(x))$ instabilities allows for substitution with stable $\texttt{log1mexp}$
>
> The first contribution is a prerequisite to the stabilizing application of $\texttt{log1mexp}$. It further enhances compatibility with neural network settings by removing the need for positivity-enforcing link functions and allowing most computation to stay in log-space.
>
> We hope you share the perspective of other reviewers regarding the significant potential impact of our KS-related contributions, despite their apparent simplicity. Reviewer 3tyx acknowledges these contributions as “quite simple [and] I do think it has important impact in the domain,” while Reviewer xx8r highlights their practicality, noting that “the primitives (expm1() and log1p()) are already present in various frameworks. This has the potential of rapid adoption in practical settings.”
>
> &nbsp;
>
> *Two New Latent Variable Models*.
>
> These architectures are NOT merely applications of the KS, but in fact completely new large-scale bounded-latent variable models. They only relate to the KS in that they achieve their best performance when combined with our stabilized KS distribution. This further motivates the attractiveness of the KS in latent variable modeling.
>
> - Variational Bandit Encoders (VBE) address contextual multi-armed bandits, achieving state-of-the-art performance (outperforming LMC-TS) in a fraction of the run-time, while being simple, scalable, and capable of incorporating prior knowledge.  It represents the first such contextual bandit approach to perform direct variational modeling of the expected reward posterior, and leverages the reparameterization trick for efficient large-scale training. We expect this to have broad impact due to the importance of multi-armed bandits in, e.g. recommendation systems, and clinical trials.
> - Variational Edge Encoders (VEE) are a new method for link prediction in GNNs, leveraging bounded latent variables for improved uncertainty quantification.
>
>
> &nbsp;
>
> We suspect the recognized contribution of your initial assessment may be limited to the second ($\texttt{log1mexp}$ substitution) KS related contribution. To highlight the additional novelty of the VBE and VEE, in the revised manuscript we have renamed Section 4 to *“Experiments and New Variational Architectures.”*

---

> ### Author Response · Authors · 2024-11-21
> **Authors’ response to Reviewer 275N - Part 2**
>
> **Quantitatively Comparing the stable KS against the unstable KS.**
>
> The unstable KS consistently results in training failures (NaN errors) across all experimental settings (VAEs, contextual multi-armed bandits, and link prediction). As noted in Section 4 (lines 239–240): "...models using the unstable KS produce NaN errors and are therefore excluded." To avoid clutter, these failures were omitted from figures, which may have understated their significance. *Unlike a model that converges poorly but produces usable outputs, the unstable KS fails entirely, making quantitative comparisons difficult*. This clear distinction between failed training and successful stabilization underscores the importance of addressing the underlying numerical issues.
>
> Section 3 (lines 213–215) describes the numerical cascade responsible for these failures: “This underflow results in a point mass at $x=0$ … and produces infinite gradients… [which] triggers a cascade: infinite parameter values after the optimizer step and NaN activations in the next forward pass…”.
>
> These issues are not unique to our implementation but reflect a longstanding challenge with the KS distribution, as evidenced by public forum discussions ([1]–[3]) and prior research ([4]–[7]). Such prior research ([4]–[7]) rely on parameter clamping $(a_{\text{min}}, a_{\text{max}}),  (b_{\text{min}}, b_{\text{max}})$ and uniform base distribution constraints $(u_{\text{min}}, u_{\text{max}})$ to avoid instability. While feasible for small-scale models ($\approx 10^1−10^2$ KS latent variables, as in [4]–[7]), these methods are impractical for the large-scale settings addressed in this work ($10^7$ latent variables), where an instability in a single KS latent can cause training failure.
>
> Our stabilization approach directly resolves these numerical issues, eliminating the need for such hyperparameter tuning and enabling stable training at scale. By ensuring robust computation, our method prevents catastrophic failures in large models and simplifies development workflows. This innovation has driven the ongoing integration of our stabilized KS implementation into major auto-differentiation libraries, further underscoring its practical relevance.
>
> To clarify these points, we have added *Remark 1* in Section 4 of the revised manuscript.
>
> &nbsp;
>
> **Closing Remarks**
>
> Thanks again for your thoughtful review. We have worked to address the issues raised and to clarify the full scope of our contributions. Specifically, we present a fundamental reparameterization that enables stable use of the KS distribution, along with two new architectures with significant impact in real-world applications. We also value your point on the broader applicability of this work to Dirichlet and non-parametric models, which we have highlighted further in the revised manuscript.
>
> *Having addressed your concerns, we kindly ask you to consider revising your score to reflect the work’s significantly broader contributions and relevance, which may not have been fully recognized in the initial assessment*.
>
>
> We strive to improve our paper and will be happy to continue the discussion if any outstanding issues remain.
>
> &nbsp;
>
>
> [1] https://github.com/probtorch/pytorch/pull/143#issuecomment-428211244
>
>
> [2] https://github.com/probtorch/pytorch/pull/143#issuecomment-377108639
>
>
> [3] https://github.com/pytorch/pytorch/issues/11937#issuecomment-423587839
>
>
> [4] Eric Nalisnick, Lars Hertel, and Padhraic Smyth. Approximate inference for deep latent Gaussian mixtures. In NeurIPS Worksh. on Bayes. Deep Learn., 2016.
>
>
> [5] Eric Nalisnick and Padhraic Smyth. Stick-breaking variational autoencoders. In ICLR, 2017.
>
>
> [6] Andrew Stirn, Tony Jebara, and David Knowles. A new distribution on the simplex with auto-encoding applications. In NeurIPS, 2019
>
>
> [7] Rachit Singh, Jeffrey Ling, and Finale Doshi-Velez. Structured variational autoencoders for the Beta-Bernoulli process. In NeurIPS Workshop Adv. Approx. Bayes. Infer., 2017.

---

> > ### Comment · Reviewer_275N · 2024-11-26
> >
> > After reading other reviewers' comments and the author's feedback, I decided to maintain my score.

---

### Official Review · Reviewer_3tyx · 2024-11-10

**Soundness:** 4
**Presentation:** 3
**Contribution:** 3
**Rating:** 6
**Confidence:** 3

**Summary:**

The paper  resolves numerical instabilities in the KS inverse CDF and log-pdf, improving implementations in libraries like PyTorch and TensorFlow. The authors leverage the stabilized KS distribution in scalable models, demonstrating improved exploration-exploitation trade-offs in contextual multi-armed bandits and enhanced uncertainty quantification in graph neural networks. These contributions position the stabilized KS distribution as a key tool for bounded latent variables in variational models.

**Strengths:**

The paper's strengths lie in its originality in addressing numerical instabilities in the KS distribution. While the idea is simple, resolving these issues unlocks the KS distribution's potential in applications like contextual multi-armed bandits and graph neural networks, enhancing exploration and uncertainty quantification. The work is of high quality, with validation and clear exposition, making it an impactful contribution to bounded latent variable modeling. Although the solution is quite simple, I do think it has important impact in the domain.

**Weaknesses:**

The key weakness of the paper is the lack of direct evidence quantifying the impact of numerical instability on performance outcomes in the experiments. While the authors resolve known instabilities in the KS distribution, it is unclear how these instabilities previously affected results or how the stabilization improves them. I see the empirical resutls,but I do not understand how.

**Questions:**

The figures lack sufficient information in their titles and captions to guide readers effectively. In particular, Figure 2 is unclear, as the plot and the meaning of the colors are not well-explained, making it difficult to interpret.

---

> ### Author Response · Authors · 2024-11-21
> **Authors’ response to Reviewer 3tyx - Part 1**
>
> Thank you for your thoughtful review and for recognizing the impact, originality, and quality of our work. Below, we address your primary concerns by clarifying the role of KS instability in experimental outcomes and improving figure clarity. We then highlight significant novelty and even broader impact which perhaps was overlooked in your initial assessment.
>
>
> &nbsp;
>
>
> **Clarifying the Impact of KS Instability on Experimental Outcomes.**
>
>
> The unstable KS consistently results in training failures (NaN errors) across all experimental settings (VAEs, contextual multi-armed bandits, and link prediction). As noted in Section 4 (lines 239–240): "...models using the unstable KS produce NaN errors and are therefore excluded." These failures were omitted from figures to avoid clutter but may have obscured their significance.
>
>
> Section 3 (lines 213–215) describes the numerical cascade responsible for these failures: “This underflow results in a point mass at $x=0$ … and produces infinite gradients… [which] triggers a cascade: infinite parameter values after the optimizer step and NaN activations in the next forward pass…”.
>
>
> These issues are not unique to our implementation but reflect a longstanding challenge with the KS distribution, as evidenced by public forum discussions ([1]–[3]) and prior research ([4]–[7]). Such prior research ([4]–[7]) rely on parameter clamping $(a_{\text{min}}, a_{\text{max}}),  (b_{\text{min}}, b_{\text{max}})$ and uniform base distribution constraints $(u_{\text{min}}, u_{\text{max}})$ to avoid instability. While feasible for small-scale models ($\approx 10^1−10^2$ KS latent variables, as in [4]–[7]), these methods are impractical for the large-scale settings addressed in this work ($10^7$ latent variables), where an instability in a single KS latent can cause training failure.
>
>
> Our stabilization approach directly resolves these numerical issues, eliminating the need for such hyperparameter tuning and enabling stable training at scale. By ensuring robust computation, our method prevents catastrophic failures in large models and simplifies development workflows. This innovation has driven the ongoing integration of our stabilized KS implementation into major auto-differentiation libraries, further underscoring its practical relevance.
>
>
> To clarify these points, we have added *Remark 1* in Section 4 of the revised manuscript.
>
>
> &nbsp;
>
>
> **Figure Clarity**.
>
>
> Thank you for your feedback on figure clarity. We have revised the captions for Figures $2$, $4$, and $5$ to improve the ability to guide readers through the work.

---

> ### Author Response · Authors · 2024-11-21
> **Authors’ response to Reviewer 3tyx - Part 2**
>
> **Four Primary Contributions**
>
>
> We would like to bring to your attention the four main contributions of this work which may have been overlooked in the initial assessment; these contributions are stated in the introduction and we provide significant evidence for them in the experiments.
>
>
> &nbsp;
>
>
> *Two Kumaraswamy (KS) Related Contributions*.
> - Reparameterization of KS functions: Equations 4–10 reparameterize the distribution-functions in terms of unconstrained logarithmic parameters. Their algebraic form makes the dominant $\log(1-\exp(x))$ instabilities explicit and isolated.
> - Stabilization via $\texttt{log1mexp}$: Isolating the the dominant $\log(1-\exp(x))$ instabilities allows for substitution with stable $\texttt{log1mexp}$
>
>
> The first contribution is a prerequisite to the stabilizing application of $\texttt{log1mexp}$. It further enhances compatibility with neural network settings by removing the need for positivity-enforcing link functions and allowing most computation to stay in log-space.
>
>
> &nbsp;
>
>
>
>
> *Two New Latent Variable Models*.
>
>
> These architectures are NOT merely applications of the KS, but in fact completely new large-scale bounded-latent variable models. They only relate to the KS in that they achieve their best performance when combined with our stabilized KS distribution. This further motivates the attractiveness of the KS in latent variable modeling.
>
>
> - Variational Bandit Encoders (VBE) address contextual multi-armed bandits, achieving state-of-the-art performance (outperforming LMC-TS) in a fraction of the run-time, while being simple, scalable, and capable of incorporating prior knowledge.  It represents the first such contextual bandit approach to perform direct variational modeling of the expected reward posterior, and leverages the reparameterization trick for efficient large-scale training. We expect this to have broad impact due to the importance of multi-armed bandits in, e.g. recommendation systems, and clinical trials.
> - Variational Edge Encoders (VEE) are a new method for link prediction in GNNs, leveraging bounded latent variables for improved uncertainty quantification.
>
>
> &nbsp;
>
>
> We suspect the recognized contribution of your initial assessment may be limited to the second ($\texttt{log1mexp}$ substitution) KS related contribution. To highlight the additional novelty of the VBE and VEE, in the revised manuscript we have renamed Section 4 to *“Experiments and New Variational Architectures.”*
>
>
> &nbsp;
>
>
> **Closing Remarks**
>
>
> We thank you for your thoughtful review and for acknowledging the simplicity of this work as a strength that enables impactful contributions to bounded latent variable modeling. We also appreciate your recognition of the work’s high quality and clear exposition. We hope our response has clarified the importance of resolving KS instabilities for experimental outcomes and highlighted the broader scope of our contributions: a stabilized KS distribution and two novel architectures with substantial real-world impact.
>
>
> *Having addressed your concerns, we kindly ask you to consider revising your score to reflect the work’s significantly broader contributions and impact, which may not have been fully recognized in the initial assessment*.
>
> We strive to improve our paper and will be happy to continue the discussion if any outstanding issues remain.
>
>
> &nbsp;
>
>
> [1] https://github.com/probtorch/pytorch/pull/143#issuecomment-428211244
>
>
> [2] https://github.com/probtorch/pytorch/pull/143#issuecomment-377108639
>
>
> [3] https://github.com/pytorch/pytorch/issues/11937#issuecomment-423587839
>
>
> [4] Eric Nalisnick, Lars Hertel, and Padhraic Smyth. Approximate inference for deep latent Gaussian mixtures. In NeurIPS Worksh. on Bayes. Deep Learn., 2016.
>
>
> [5] Eric Nalisnick and Padhraic Smyth. Stick-breaking variational autoencoders. In ICLR, 2017.
>
>
> [6] Andrew Stirn, Tony Jebara, and David Knowles. A new distribution on the simplex with auto-encoding applications. In NeurIPS, 2019
>
>
> [7] Rachit Singh, Jeffrey Ling, and Finale Doshi-Velez. Structured variational autoencoders for the Beta-Bernoulli process. In NeurIPS Workshop Adv. Approx. Bayes. Infer., 2017.

---

> > ### Comment · Reviewer_3tyx · 2024-11-25
> > **response**
> >
> > Thanks a lot for the response. My concerns have been addressed and I will keep the score.

---

### Public Comment · ~Mariano_Tepper1 · 2024-11-25
**On the usefulness of the stabilized Kumaraswamy distribution**

I have recently encountered this paper and found its contributions meaningful. As such, I am taking the liberty to add an external perspective.

The Kumaraswamy distribution was designed to be a simpler alternative to the Beta distribution. As the authors state, it offers many of the charastersitics thar make the Beta appealing, while being much easier to compute: it has closed-form CDF, inverse CDF, and derivative. However, simplicity comes at the cost of numerical instability, which the authors'constributions fix.

I have tried the stabilization proposed by the authors and it is easy to implement and it works like a charm. Now that it is of practical use thanks to the stabilization, I believe that many researchers and practitioners of ML and statistics would find this distribution very useful but, until today, it is not well-known. This paper can help disseminate the idea that a new powerful, computationally efficient, and stable tool is available, and the whole community will benefit from it.

---

### Note · Authors · 2024-12-02

I have read and agree with the venue's withdrawal policy on behalf of myself and my co-authors.